# ScalaFlux: A scalable approach to quantify fluxes in metabolic subnetworks

**Pierre Millard**[1], **Uwe Schmitt**[2], **Patrick Kiefer**[3], **Julia A. Vorholt**[3], **Stéphanie Heux**[1], **Jean-Charles Portais**[1,4,5]\*

**1** TBI, Université de Toulouse, CNRS, INRAE, INSA, Toulouse, France, **2** Scientific IT Services, ETH Zurich, Zurich, Switzerland, **3** Institute of Microbiology, Department of Biology, ETH Zurich, Zurich, Switzerland, **4** MetaboHUB-MetaToul, National infrastructure of metabolomics and fluxomics, Toulouse, France, **5** STROMALab, Université de Toulouse, INSERM U1031, EFS, INP-ENVT, UPS, Toulouse, France

\* jean-charles.portais@insa-toulouse.fr

**Data Availability Statement:** All relevant data are within the manuscript and its Supporting Information files.

**Funding:** J-CP, SH, and PM were supported by French National Research Agency under grant

## Abstract

$^{13}$C-metabolic flux analysis ($^{13}$C-MFA) allows metabolic fluxes to be quantified in living organisms and is a major tool in biotechnology and systems biology. Current $^{13}$C-MFA approaches model label propagation starting from the extracellular $^{13}$C-labeled nutrient(s), which limits their applicability to the analysis of pathways close to this metabolic entry point. Here, we propose a new approach to quantify fluxes through any metabolic subnetwork of interest by modeling label propagation directly from the metabolic precursor(s) of this subnetwork. The flux calculations are thus purely based on information from within the subnetwork of interest, and no additional knowledge about the surrounding network (such as atom transitions in upstream reactions or the labeling of the extracellular nutrient) is required. This approach, termed ScalaFlux for SCALAble metabolic FLUX analysis, can be scaled up from individual reactions to pathways to sets of pathways. ScalaFlux has several benefits compared with current $^{13}$C-MFA approaches: greater network coverage, lower data requirements, independence from cell physiology, robustness to gaps in data and network information, better computational efficiency, applicability to rich media, and enhanced flux identifiability. We validated ScalaFlux using a theoretical network and simulated data. We also used the approach to quantify fluxes through the prenyl pyrophosphate pathway of *Saccharomyces cerevisiae* mutants engineered to produce phytoene, using a dataset for which fluxes could not be calculated using existing approaches. A broad range of metabolic systems can be targeted with minimal cost and effort, making ScalaFlux a valuable tool for the analysis of metabolic fluxes.

## Author summary

Metabolism is a fundamental biochemical process that enables all organisms to operate and grow by converting nutrients into energy and 'building blocks'. Metabolic flux analysis allows the quantification of metabolic fluxes *in vivo*, i.e. the actual rates of biochemical conversions in biological systems, and is increasingly used to probe metabolic activity in biology, biotechnology and medicine. Isotope labeling experiments coupled with

Enzinvivo (ANR-16-CE11-0022). J-CP benefited from a temporary full-time researcher position funded by INSERM. PK, US, and JAV were funded by ETH Zurich. The funders had no role in study design, data collection and analysis, decision to publish, or preparation of the manuscript.The funders had no role in study design, data collection and analysis, decision to publish, or preparation of the manuscript.

**Competing interests:** The authors have declared that no competing interests exist.

mathematical models of large metabolic networks are the most commonly used approaches to quantify fluxes within cells. However, many biological questions only require flux information from a subset of reactions, not the full network. Here, we propose a new approach with three main advantages over existing methods: better scalability (fluxes can be measured through a single reaction, a metabolic pathway or a set of pathways of interest), better robustness to missing data and information gaps, and lower requirements in terms of measurements and computational resources. We validate our method both theoretically and experimentally. ScalaFlux can be used for high-throughput flux measurements in virtually any metabolic system and paves the way to the analysis of dynamic fluxome rearrangements.

This is a *PLOS Computational Biology* Methods paper.

## Introduction

Metabolic flux analysis (MFA) with stable isotope tracers, typically a $^{13}$C-labeled carbon source, allows intracellular fluxes to be quantified in a wide range of organisms and is now a major tool in the fields of biotechnology [1–3], systems biology [4–6] and medicine [7,8]. Current approaches rely on isotopic models to simulate tracer propagation through metabolic networks in (pseudo) steady-state condition [1,9–14]. Fluxes are then estimated by fitting experimental concentrations and isotopic profiles of metabolites. Current simulation frameworks require known and constant label input(s). The only constant label input(s) is (are) the isotopically-labeled nutrient(s) in the extracellular medium, which must therefore be included in the flux model. In practice, this means that all metabolic models must explicitly include the labeled nutrient(s) initially provided to the biological system and all the pathways that distribute the isotopic tracer up to the pathway of interest. To ensure fluxes are identifiable, the extracellular fluxes and the labeling of upstream metabolites must also be measured (as well as the intracellular metabolite concentrations for instationary $^{13}$C-MFA approaches). This is a major limitation for investigating i) pathways far downstream of the labeled nutrient(s), ii) networks with reaction gaps (e.g. an uncertain network topology), iii) incomplete datasets, iv) experiments performed in rich media, or v) situations where the isotopic transitions remain uncertain or complex (e.g. $^{2}$H tracer) [1,15]. This also makes the entire experimental and computational workflow very time consuming, costly and error prone. Overall, the modeling requirement that the tracer has to be propagated right from the extracellular nutrient limits the application of flux measurements to pathways closely related to the label input. The vast majority of existing $^{13}$C-flux studies focus indeed on central carbon metabolism, and most $^{15}$N-flux studies focus on the nitrogen assimilation network [1,4,6,16,17]. Alternative $^{13}$C-MFA frameworks such as metabolic flux ratio analysis [18,19] and kinetic flux profiling [16] were developed, but they are far to be generic since they are limited to the analysis of a few topological motifs based exclusively on mass spectrometry (MS) data. There is therefore a need for more robust and scalable approaches to quantify metabolic fluxes in biochemical systems.

Here, we propose a new isotope-based-MFA approach, named ScalaFlux, to measure fluxes at the level of any metabolic subnetwork of interest, in which label propagation is modeled directly from the metabolic precursor(s) of this subnetwork. ScalaFlux uses a limited amount of input data and increases the number of pathways that can be accessed, while significantly

reducing experimental and computational requirements. We demonstrate the value of Scala-Flux with *in silico* simulations and its practical applicability by quantifying *in vivo* fluxes in the yeast prenyl pyrophosphate pathway.

## Results

### Basic principle: Reconsidering label inputs

Understanding the basic principle of the proposed approach requires some concepts and terminology that are introduced and illustrated using the example network shown in Fig 1. This network of 18 metabolites and 20 reactions includes three topological motifs classically found in metabolism: a linear pathway, a branching node and a cycle. We refer to the initial source(s) of label–i.e. the extracellular nutrient(s), here $X_{out}$–as the *global label input(s)* for the metabolic network. After $X_{out}$ is switched from natural abundance to isotopically labeled, the isotopic tracer propagates through the metabolic network and the intracellular metabolites ($X_{in}$, $A$, . . ., $O$), which are progressively labeled as a function of metabolite concentrations and fluxes. Fluxes can then be estimated using a model-based approach by minimizing the difference between experimental labeling data and the labeling profiles simulated by the model.

Current non-stationary $^{13}$C-flux calculation frameworks require constant label input(s) so the global label input(s) must be included in the flux models. To specifically measure the flux through reaction r16 in the example network, the flux model (red boundaries in Fig 1A) must contain $X_{out}$ and all the reactions that contribute to isotope propagation up to the product of r16. This flux model includes a total of 17 reactions, 1 (global) label input and 14 metabolic intermediates (Fig 1B). Measurements of metabolite concentrations and labeling at several

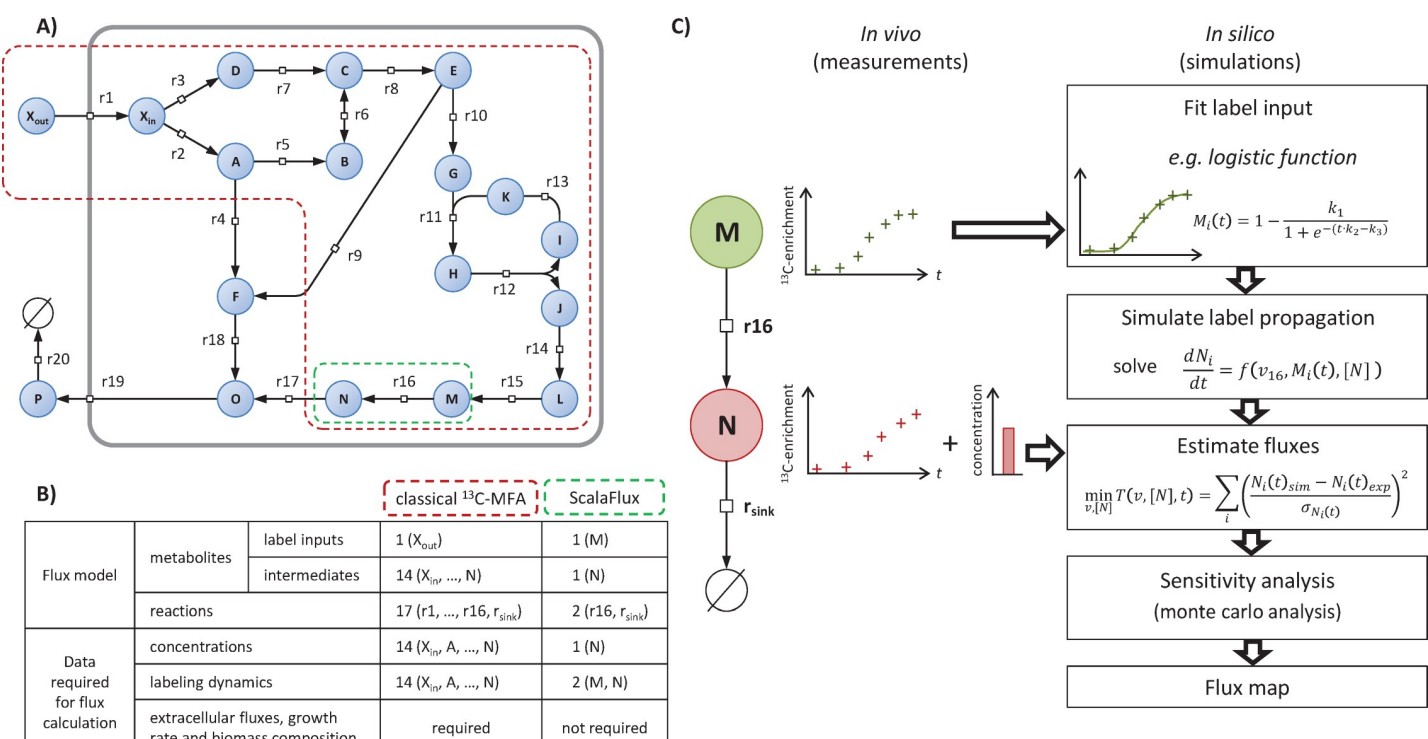

| | | | classical $^{13}$C-MFA | ScalaFlux |
|---|---|---|---|---|
| **Flux model** | metabolites | label inputs | 1 ($X_{out}$) | 1 (M) |
| | | intermediates | 14 ($X_{in}$, ..., N) | 1 (N) |
| | reactions | | 17 (r1, ..., r16, $r_{sink}$) | 2 (r16, $r_{sink}$) |
| **Data required for flux calculation** | concentrations | | 14 ($X_{in}$, A, ..., N) | 1 (N) |
| | labeling dynamics | | 14 ($X_{in}$, A, ..., N) | 2 (M, N) |
| | extracellular fluxes, growth rate and biomass composition | | required | not required |

**Fig 1. Principle of ScalaFlux.** Panel **A** shows an example network in the Systems Biology Graphical Notation format (SBGN, www.sbgn.org) [20] to illustrate the basic principle of ScalaFlux. The flux models (and associated datasets) required to quantify the flux through reaction r16 using classical non-stationary $^{13}$C-MFA and ScalaFlux are compared in panel **B**. The ScalaFlux model, the set of measurements required for the flux calculation, and the flux calculation workflow are shown in panel **C**.

nodes of the network as well as of extracellular fluxes and biomass composition are required to calculate the flux.

We propose a more scalable $^{13}$C-flux approach, named ScalaFlux for SCALAble metabolic FLUX analysis, to quantify fluxes through a subnetwork of interest using internal information from this network only. ScalaFlux does not require data on the extracellular labeled nutrient, upstream metabolites, or any other knowledge about the surrounding network. The flux model encodes a metabolic subsystem (i.e. a subset of the cellular metabolic network) and specifically contains the reaction(s) of interest, as illustrated in Fig 1 and described in detail below. All the metabolic substrates in this subsystem are considered *local label inputs*, and label propagation is simulated directly from these local label inputs. If the reaction of interest is r16, the labeling dynamics of *M* is defined as the local label input of the corresponding subsystem to simulate the labeling dynamics of *N*. In contrast to global label inputs, which are constant, known and controlled, the labeling of local label inputs changes with time, is not known *a priori* and cannot be controlled. Label incorporation can nevertheless be determined experimentally and be used for the downstream reactions. Using these discrete measurements as direct label inputs for simulations would result in sharp changes in label input at each measurement time and thereby yield stiff equations and simulation artifacts. The first step of the ScalaFlux workflow (Fig 1C) therefore consists in transforming the discrete measurements into a continuous (time-dependent) representation by fitting analytical functions, ensuring smooth variations as a function of time. A system of ordinary differential equations (ODEs) can then be constructed using conventional frameworks to simulate label propagation from the local label input(s). By combining this simulation approach with optimization routines, fluxes can be estimated by fitting experimental data. This workflow has been implemented in a major update of IsoSim [21] (see Methods for details).

Importantly, the studied subsystem can include larger parts of the network, as detailed in the following sections. This means that any given (set of) flux(es) can be quantified independently of the rest of the metabolic network, with no additional measurements (extracellular fluxes, growth rates, biomass composition, concentrations and labeling of upstream metabolites), and independently of the (often incomplete) knowledge of the metabolic network outside the boundaries of the subsystem under study.

ScalaFlux exploits many concepts from non-stationary $^{13}$C-MFA and thus benefits directly from recent advances in the field, such as efficient mathematical frameworks for experimental design [13,14,22–24], simulation [14,25–27], optimization [10,28] and sensitivity analysis [14,29]. Because it is based on detailed modeling of isotope propagation, ScalaFlux is generic with respect to the network topology (flux models can include branching nodes, cycles, or any other of the topological motifs that compose metabolic networks), the isotopic tracer ($^2$H, $^{13}$C, $^{15}$N, etc), and the type of isotopic measurement (MS, MS/MS, NMR, etc). The flux analyses presented in the rest of the article are based on mean molecular enrichment data collected by mass spectrometry in $^{13}$C-labeling experiments.

## Construction of flux models

Flux models must precisely describe the topology of the subnetwork of interest while ensuring independence from the surrounding network. A generic procedure is presented in this section to streamline the construction of self-consistent flux models of any part of a metabolic network.

We define a *minimal subsystem* $S_Y$ as the minimal set of reactions required to simulate the labeling dynamics of a given metabolite *Y*. A metabolic network containing *n* metabolic intermediates can thus be decomposed into *n* minimal subsystems. The minimal subsystem $S_Y$

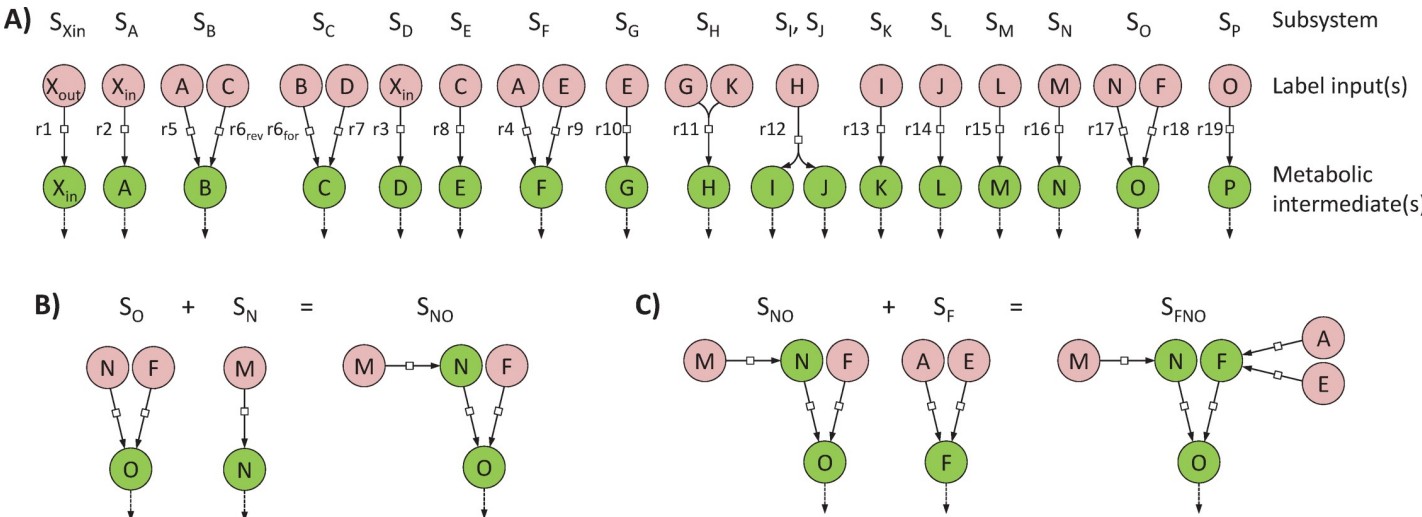

**Fig 2. Network decomposition to construct flux models.** The metabolic network shown in Fig 1A can be decomposed into 17 minimal subsystems (panel **A**) which are sufficient to simulate the labeling dynamics of metabolic intermediates (green circles) from the local label input(s) (red circles). Each minimal subsystem is self-consistent and can be used for independent flux calculations. These minimal subsystems can also be combined to analyze larger subsystems, as shown in panels **B** and **C**.

must include all the reactions that produce $Y$ (since they may all affect its labeling dynamics), with their substrates corresponding to local label inputs. For practical modeling reasons, a sink reaction consuming $Y$ has to be included to avoid its accumulation, in keeping with the metabolic steady-state assumption (i.e. metabolite concentrations are constant). Each minimal subsystem is self-consistent and can be incorporated into a flux model to estimate fluxes through the included reactions. This modular representation is the essence of the scalability of ScalaFlux. We used this procedure to decompose the example network shown in Fig 1A into 17 minimal subsystems, as shown in Fig 2A. Note that reaction r6, which is reversible, is present in two subsystems ($S_B$ and $S_C$) to account for its forward and reverse fluxes [21].

To analyze larger subnetworks that include several reactions of interests, the individual minimal subsystems that compose this subnetwork should be combined (Fig 2). Two subsystems can be combined when they share a common metabolite, e.g. the two minimal subsystems $S_Y$ and $S_Z$ can be merged if $Y$ is a local label input of $Z$. The local label inputs of the resulting subsystem $S_{YZ}$ are all the local label inputs except $Y$, which is now an intermediate of $S_{YZ}$. This ensures that all the reactions (and local label inputs) that contribute to the labeling dynamics of $Y$ and $Z$ are included. For instance, to quantify fluxes through the set of reactions {r4, r9, r16, r17, r18} in the example network, $S_O$ can first be united with $S_N$ (since the metabolic intermediate $N$ is a local label input of $S_O$) (Fig 2B), and the resulting subsystem $S_{NO}$ can then be merged with $S_F$ (Fig 2C). The final subsystem $S_{FNO}$ contains all the reactions of interest and has three local label inputs ($A$, $E$ and $M$) and three intermediates ($F$, $N$ and $O$).

### Flux calculation in minimal metabolic subsystems

The minimal set of measurements required to estimate fluxes in a minimal subsystem $S_Y$ consists of i) the labeling dynamics of its local label input(s) (used to simulate tracer propagation) and ii) the labeling dynamics of $Y$ (used for flux estimation). These transient label dynamics are thus sufficient to estimate the turnover rate of $Y$, i.e. the ratio between its pool and its biosynthetic flux. In a branched pathway, this information is also sufficient to determine the contribution of each converging reaction to the biosynthesis of $Y$. Absolute fluxes can be

estimated when the absolute concentration of $Y$ is available. The absolute *in vivo* flux through a given reaction in any linear pathway can thus be estimated from reactant data alone.

To ensure flux identifiability, the isotopic information collected on a given (sub)set of carbon atom(s) of the label input(s) must match the isotopic information collected on the corresponding (sub)set of carbon atom(s) of the product(s). For the simplest minimal subsystem containing a single unimolecular reaction (e.g. $S_A$), carbon atoms of the label input ($X_{in}$) directly matches carbon atoms of the product ($A$). The flux (r2) can thus be calculated from global information collected on each metabolite (isotopologue distributions or mean molecular enrichments measured by MS) as well as from positional information on matching atoms (specific enrichments or positional isotopomers measured by NMR). For minimal subsystems involving condensation reactions (e.g. $S_H$), the flux (r11) can be estimated from global measurements of the labeling dynamics of the two label inputs ($G$ and $K$) and of the product ($H$). The labeling dynamics of only one of the two label inputs is sufficient for flux calculation when the labeling of the corresponding subset of carbon atoms of the product is available (e.g. when a fragment containing these atoms can be measured by MS/MS, or when positional information on matching atoms are accessible). Finally, for minimal subsystems where the product is formed by a cleavage reaction (e.g. $S_I$), global isotopic information may be collected on the product ($I$), and the corresponding information must be measured on a matching fragment of the local label input ($H$). If no matching fragment is available, fluxes may still be calculated using positional information collected by NMR. As an alternative strategy, identifiability may also be enhanced by combining minimal subsystems (see section "*From individual reactions to metabolic pathways*: *combining minimal subsystems enhances flux identifiability and precision*").

ScalaFlux was tested on the metabolic network shown in Fig 1A. Metabolite concentrations and fluxes were initialized at the values listed in the Supporting information (S1 Table), and label propagation through this network was simulated to create a theoretical dataset (S1 Fig). We estimated fluxes in all minimal subsystems (Fig 3A) from these theoretical labeling dynamics. The transient $^{13}$C-enrichments of all local label inputs were accurately described by fitting a double logistic function (S2 Fig), and these analytical functions were used as label inputs for flux calculation.

The labeling dynamics of metabolic intermediates are accurately fitted by the flux models for all minimal subsystems (S3 Fig). The estimated fluxes are in good agreement with the true values used to run the simulations ($R^2 = 0.98$, Fig 3B), with an average relative error of 7%. For the reversible reaction r6, both the forward and reverse reaction rates were determined.

We tested how the shape of the functions that represent local label inputs affect the fluxes by degrading artificially the quality of the fit of label inputs. We varied parameters of the analytical functions (parameters were sampled randomly within ± 5% of their optimal values), and we analyzed the distribution of errors on the estimated fluxes as function of the error introduced on the representation of label inputs (S4 Fig). We carried out this analysis on two minimal subsystems composed of a single reaction ($S_N$) or of two converging reactions ($S_F$). For the minimal subsystem $S_N$, the error on the estimated flux r16 was minimal for parameters corresponding to the best fit of label inputs, and the error increased when the fit was degraded (S4 Fig). The same phenomenon was observed in $S_F$, both for individual fluxes (r4 and r9) and for the relative contribution of each reaction to the biosynthesis of $F$ (S4 Fig). The flux sensitivity depends on the subsystem, on the reaction (e.g. the flux r16 in $S_N$ is more sensitive against parameter variation than the flux r4 or r9 in $S_F$), and on the inferred information (e.g. for $S_F$, the flux ratio seems to be less sensitive against parameter variation than the flux r9). These results stress the need of providing accurate representations of label inputs for flux calculation. If a given analytical function cannot fit accurately the experimental labeling dynamics of some

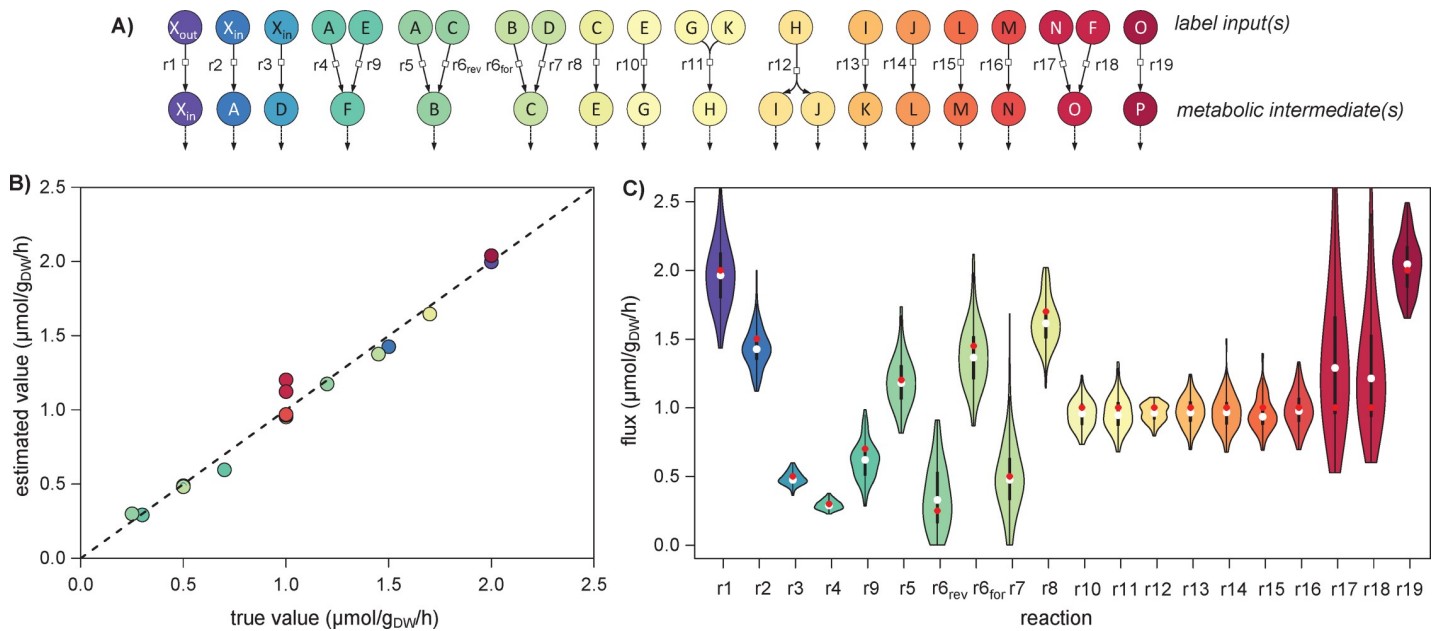

**Fig 3. Fluxes through each reaction of the example network (Fig 1A) estimated by analyzing all minimal subsystems.** Fluxes were estimated independently in all the minimal subsystems shown in panel **A**. The estimated fluxes are in good agreement with the true values ($R^2 = 0.98$, p-value $= 1.10^{-14}$, panel **B**). The distribution of fluxes estimated from 200 noisy datasets are shown in panel **C**, with the true value used for simulation shown as a red dot and the median of the estimated fluxes shown as a white dot.

label inputs, one should therefore find another function that better represents label inputs, and test how the shape of this function impacts their own flux calculations.

The robustness of ScalaFlux to measurement noise was assessed by a Monte Carlo sensitivity analysis [29]. Fluxes were estimated from 200 datasets in which Gaussian noise was added to the theoretical data, assuming a typical relative standard deviation 2% for $^{13}$C-enrichments and of 10% for concentrations [30,31]. The distribution of fluxes estimated from these datasets indicates that the precision of the method is good, with an average relative standard deviation of 13% (Fig 3C). All flux confidence intervals include the true flux values used for simulations. ScalaFlux is thus robust to measurement uncertainty.

Overall, the proposed approach provides accurate estimates of absolute fluxes, with no measurement of extracellular uptake or production fluxes having been provided as input. This proof of concept example validates the proposed approach.

### From individual reactions to metabolic pathways: Combining minimal subsystems enhances flux identifiability and precision

As well as quantifying fluxes in minimal subsystems, ScalaFlux can be used to analyze larger subsystems. Just like in minimal subsystems, the set of measurements required to estimate fluxes in larger subsystems consists of i) the labeling dynamics of local label input(s) and ii) the labeling dynamics of (at least one) metabolic intermediate(s).

To illustrate the value of this scalability, we explored different options to estimate the flux through the pathway composed of the seven reactions {r10, . . ., r16} (Fig 4A). We identified a total of 29 subsystems (and associated datasets, Fig 4B) that potentially enable flux evaluation through this pathway. Of course, this flux can be estimated through each reaction individually, as demonstrated above, corresponding to subsystems $S_G$, $S_H$, $S_{IJ}$, $S_K$, $S_L$, $S_M$, and $S_N$ in Fig 4B. Several reactions in this pathway can also be combined into a single flux model (following the

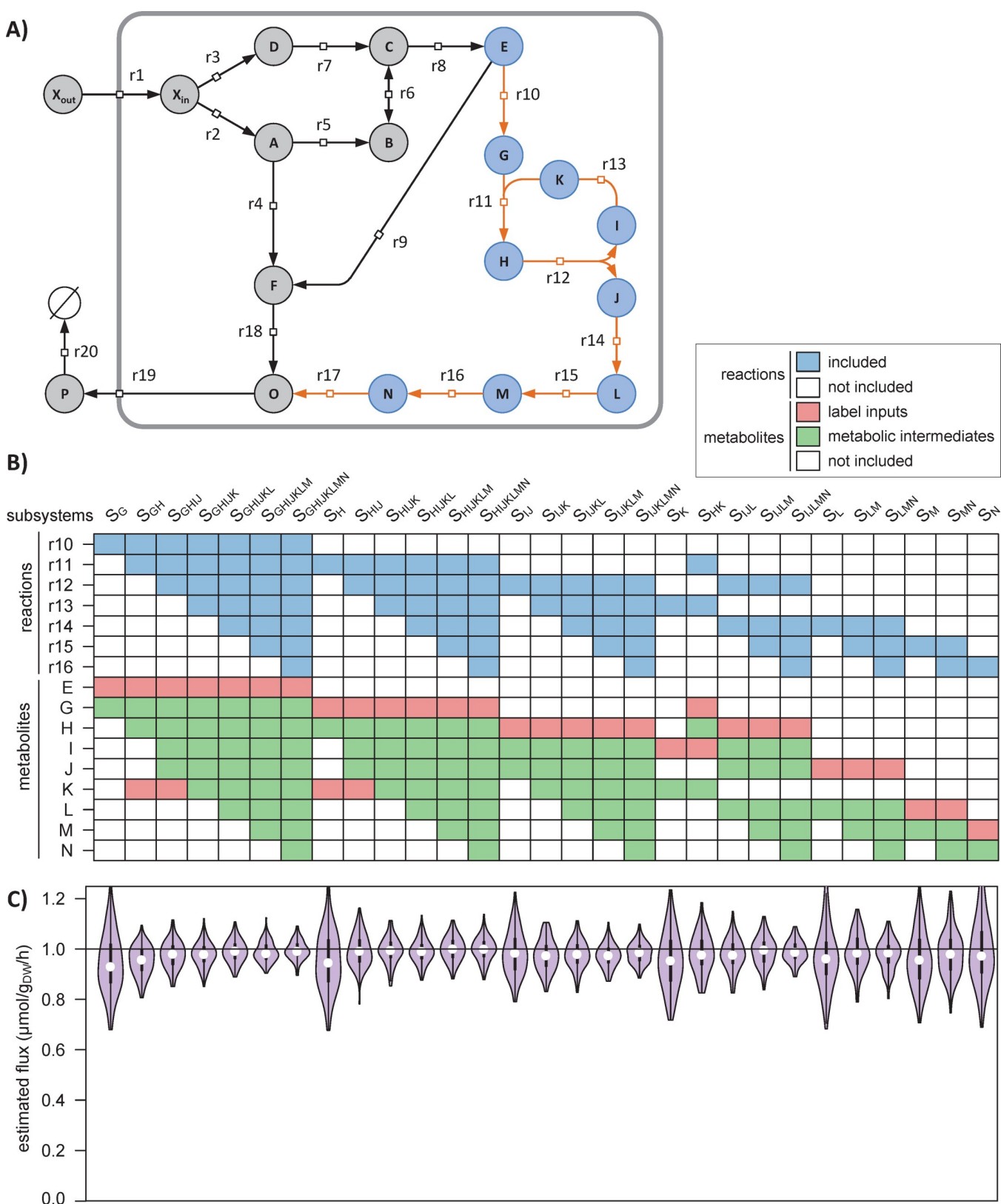

**Fig 4. Demonstration of the scalability of ScalaFlux.** The absolute flux through the pathway r10-r16 (orange reactions in panel **A**) can be quantified in 29 different subsystems (columns in panel **B**), each of which i) include different reactions (in blue) and ii) exploit different sets of measurements (labeling of local

label inputs in red, and concentrations and labeling of metabolic intermediates in green). The fluxes estimated for each subsystem are shown in panel **C** and are compared to the true value (1.0 μmol/g$_{DW}$/h, horizontal line).

rules defined in section *Construction of flux models*), e.g. by merging two connected subsystems as done for subsystems $S_{GH}$, $S_{HK}$, $S_{LM}$ and $S_{MN}$.

The fluxes calculated for each subsystem are shown in Fig 4C and are all close to the true value (1.0 μmol/g$_{DW}$/h). Increasing the size of the subsystem used for flux calculation improves both the accuracy and precision of the estimated fluxes. For instance, the flux was estimated at 0.96±0.10 μmol/g$_{DW}$/h for the minimal subsystem $S_G$ and at 0.99±0.04 μmol/g$_{DW}$/h for the largest subsystem $S_{GHIJKLM}$. This is because the reconciliation of larger datasets during flux calculation increases the robustness of the approach. Experimental and analytical efforts can thus be optimized depending on the required flux precision.

Another advantage of this scalability is that it increases flux identifiability. For instance, estimating the flux through r16 is possible via the flux model of $S_N$, provided the labeling dynamics of *M* is available (Fig 4B). If *M* cannot be measured, label propagation cannot be simulated and no flux can be estimated. However, if the labeling dynamics of *L* is available, the flux through *r16* can still be estimated using the flux model of $S_{MN}$ for which the labeling dynamics of the local label input *L* is known. The most appropriate flux model can thus be selected based on the available data, without making the additional assumptions or oversimplifications required by current approaches (e.g. using hypothetical tracer atom transitions from upstream pathways, defining reversible reactions as irreversible, or lumping reactions). Since each subsystem can be investigated independently of the rest of the cellular network, poorly identified parts of the network (e.g. due to missing measurements or an uncertain topology) do not affect the reaction(s) of interest.

## Biosynthesis of prenyl pyrophosphates in yeast

ScalaFlux provides the opportunity to reconsider published datasets from which fluxes could not be calculated because of the lack of an appropriate modeling framework. As an example application, we analyzed a published dataset on the metabolism of prenyl pyrophosphates, the precursors of isoprenoids, in the yeast *Saccharomyces cerevisiae* [32]. Isoprenoid biosynthesis starts with isopentenyl pyrophosphate (IPP), which is isomerized into dimethylallyl pyrophosphate (DMAPP) (Fig 5A). DMAPP is then condensed with another IPP to generate geranyl pyrophosphate (GPP). Longer prenyl pyrophosphates are built by successive condensation of IPP onto each intermediate, giving farnesyl pyrophosphate (FPP) from GPP and geranylgeranyl pyrophosphate (GGPP) from FPP.

The published dataset contains i) steady-state concentrations of three prenyl pyrophosphate intermediates (GPP, FPP and GGPP) measured during exponential growth on glucose and ii) 44 transient $^{13}$C-enrichments following a switch from unlabeled to U-$^{13}$C-glucose (11 time points for GPP, FPP, GGPP, and combined pools of IPP and DMAPP). These data were collected in three different strains designed to enhance phytoene production. The GGPP pool of the wild-type (WT) metabolic chassis was first increased by constructing the strain S037, which overexpresses GPP and FPP synthase (ERG20) and GGPP synthase (CrtE). A heterologous phytoene synthase (CrtB from *Pantoea ananatis*) was then expressed to convert GGPP into phytoene in strain S023. The pools of all intermediates were higher in strains S037 and S023 compared to wild type, suggesting higher fluxes, but this could not be verified because fluxes could not be inferred solely from these data. We therefore used ScalaFlux to estimate the *in vivo* flux through the prenyl pyrophosphate pathway in the three strains.

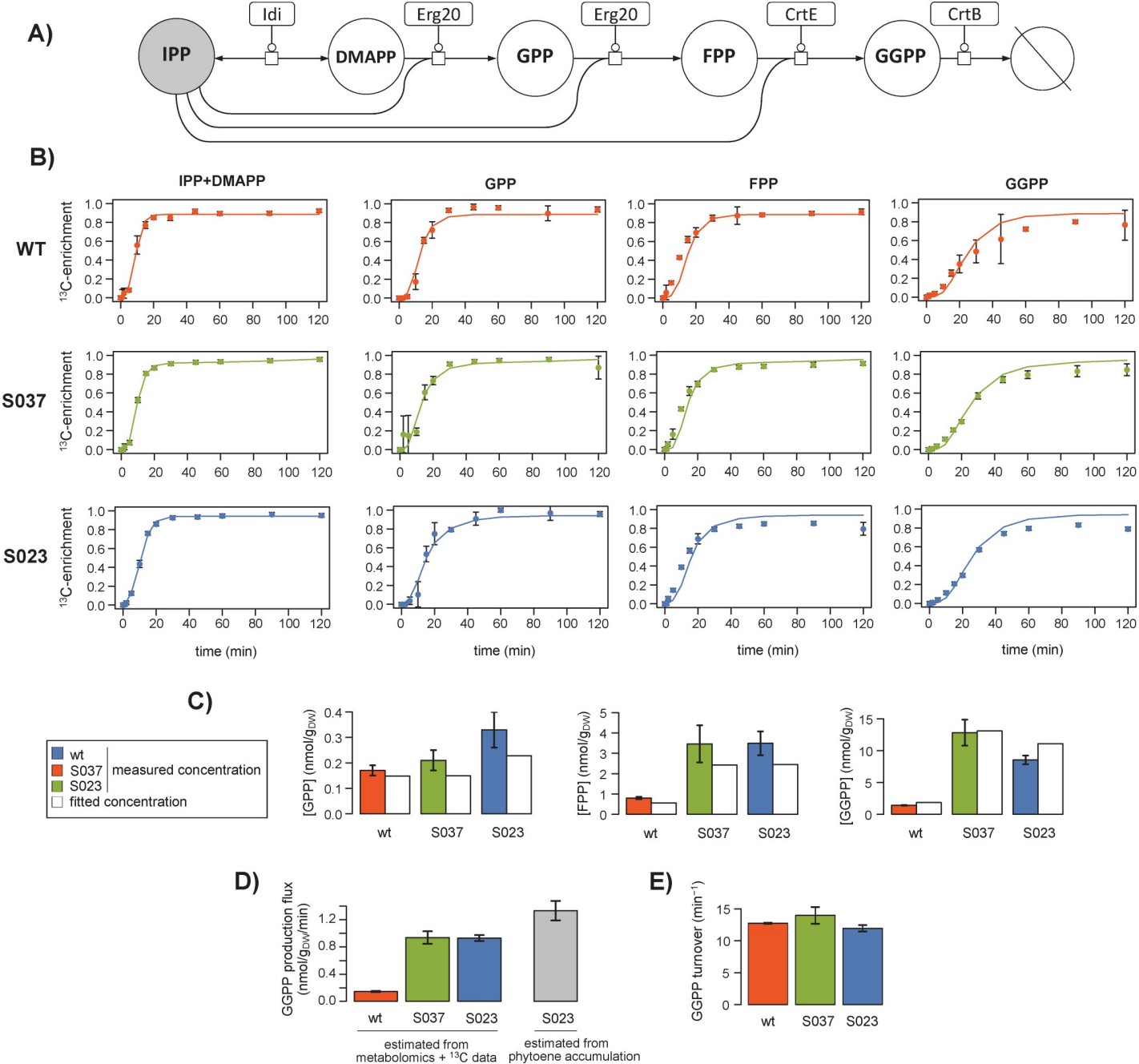

**Fig 5. $^{13}$C-metabolic flux analysis of prenyl pyrophosphate biosynthesis in *Saccharomyces cerevisiae* (wild type, S037 and S023 strains).** The yeast prenyl pyrophosphate pathway contains five reactions for the successive condensation of IPP (in grey) onto each intermediate (DMAPP, FPP, GPP and GGPP) (**A**). The labeling dynamics of IPP were fitted with a double logistic function, which was used as the local label input. Fluxes were estimated by fitting the metabolite concentrations and transient $^{13}$C-enrichments of GPP, FPP and GGPP. Experimental and fitted data are shown for each strain in panel **B** for the labeling dynamics (dots: experimental values; lines: best fit) and in panel **C** for the metabolite concentrations. The fluxes estimated in each strain are given with their standard deviations in panel **D**. The GGPP demand calculated from phytoene accumulation in strain S023 is shown in grey for comparison. The GGPP turnover rate estimated in each strain is shown in panel **E**.

The flux model is centered on the specific pathway of interest and thus only includes the five reactions shown in Fig 5A. We used a double logistic function to fit the transient labeling dynamics of IPP (i.e. mean molecular $^{13}$C-enrichment), from which accurate analytical

representations were obtained (Fig 5B). This function was used as label input to estimate fluxes by fitting the concentrations and dynamic $^{13}$C-enrichments of three other intermediates (GPP, FPP and GGPP). The good agreement between simulations and measurements (Fig 5B and 5C, $R^2 > 0.98$) indicates that the concentrations and isotopic data are consistent with the topology defined in the model. In wild-type *Saccharomyces cerevisiae*, the GGPP biosynthetic flux was estimated at 0.15±0.01 nmol/$g_{DW}$/min during exponential growth on glucose (Fig 5D). It increased to 0.94±0.08 nmol/$g_{DW}$/min in strain S037, hence confirming the relevance of the strain design strategy in improving the availability of GGPP, the precursor of phytoene biosynthesis. The flux was similar in the phytoene producing strain S023 (0.93±0.04 nmol/$g_{DW}$/min). This indicates that the increased demand for GGPP does not propagate upstream and does not affect its production, in agreement with the low reversibility of the prenyl transferase reactions. Importantly, we verified that the flux estimated by ScalaFlux in S023 was consistent with the GGPP demand for phytoene synthesis estimated from phytoene accumulation (1.33±0.16 nmol/$g_{DW}$/min, Fig 5D). The good agreement between these two independent methods demonstrates that ScalaFlux provides accurate flux measurements from datasets collected on just a few metabolic intermediates.

Finally, while qualitative interpretations suggested that the turnover rate of GGPP was stable in the different strains [32], this could not be verified because the fluxes could not be estimated. We therefore evaluated this hypothesis by calculating the GGPP turnover from the estimated fluxes and metabolite concentrations. Results indicate that GGPP turnover (Fig 5E) is indeed very similar in the three strains (WT: 12.7±0.1, S037: 14.0±1.3, S023: 11.9±0.5 min$^{-1}$), and thus confirm quantitatively that the GGPP pool increases roughly proportionally to its biosynthetic flux.

## Discussion

In current $^{13}$C-MFA approaches, label propagation has to be modeled starting from the extracellular nutrient(s), which limits their applicability to flux analysis of pathways close to this nutrient. Here, we present a novel MFA framework to investigate any reaction or set of reactions in a subnetwork of interest based on just a few targeted measurements in this subnetwork.

The scalability of ScalaFlux stems from the modular decomposition of metabolic networks into minimal subsystems, which can be analyzed independently or merged together to analyze larger subnetworks, as demonstrated using a theoretical network and simulated data. The guidelines provided to decompose a network into minimal subsystems enable intuitive reasoning and facilitate experimental design (e.g. in terms of the measurements to perform), which can be supported further by *in silico* simulations. It is important to note that flux identifiability depends on the experimental setup used (e.g. type of isotopic data, accessible measurements, sampling frequency) and on biological constraints (e.g. network topology, fluxes). We refer to previous work [12–14,22,24,33] for extensive discussion on these topics.

We validated the practical applicability of ScalaFlux by reanalyzing a published dataset on the metabolism of prenyl pyrophosphates, from which fluxes could not be calculated using current MFA approaches. Indeed, GGPP is continuously used by different processes (such as protein geranylgeranylation and membrane biosynthesis) and does not accumulate in cells. Its biosynthetic flux cannot therefore be measured *in vivo* without using isotopic tracers. Moreover, measuring this flux using stationary $^{13}$C-MFA approaches would have been impossible because of the topology of the prenyl pyrophosphate pathway. Non-stationary $^{13}$C-MFA approaches could have been used, but at much higher analytical and computational costs. The underlying model would have had to include many additional reactions involved in $^{13}$C label

propagation from glucose up to IPP, i.e. at least some of the central metabolic pathways that contribute to the labeling of acetylCoA (glycolysis, the pentose phosphate pathway, and possibly anaplerotic reactions and the TCA cycle), and the entire mevalonate pathway that produces IPP from acetylCoA. This model would thus have contained several dozen reactions, for which the associated fluxes would have had to be estimated. Our approach significantly reduces the size of the model and the number of free parameters, and thereby the computational cost of the flux calculation. Moreover, the absolute pathway flux was estimated using the metabolite concentrations and $^{13}$C-enrichments collected for just a few metabolites using a single LC-HRMS platform [32]. Using traditional approaches, the full model would have been undetermined—and no flux could have been estimated—without additional experimental data on key points in the upstream pathways (e.g. the glucose uptake flux, and the pools and transient $^{13}$C-enrichments of upper intermediates), collected with different sampling times, and analyzed with different analytical platforms. Our approach thus also reduces experimental costs and processing efforts.

ScalaFlux is fundamentally scalable, providing several different ways to quantify a given (set of) flux(es). The most appropriate flux model should be selected based on the biological question to be addressed (e.g. in terms of the fluxes to be measured or the required flux precision) and practical constraints (e.g. network knowledge or available data). For instance, fluxes through individual reactions in a linear pathway can be estimated independently using different datasets. ScalaFlux can thus potentially verify (or disprove) assumptions that are usually made in $^{13}$C-MFA (e.g. that all the reactions in a given linear pathway carry the same flux) and to identify gaps in the current knowledge (e.g. that an intermediate of an apparently linear pathway is actually consumed by another unknown reaction, or that the assumed network topology is not sufficient to explain the labeling dynamics of some of the intermediates).

ScalaFlux is also highly versatile in terms of the pathways that can be monitored. It can be used to measure fluxes through virtually any metabolic subsystem of interest: a single reaction, a pathway, or larger networks. Because it exploits concepts from non-stationary $^{13}$C-MFA, ScalaFlux can be used to investigate $C_1$-metabolism (e.g. $CO_2$ fixation, methylotrophy, folate metabolism). It also allows the quantification of metabolic fluxes that are currently difficult to measure, e.g. in secondary metabolism (such as prenyl pyrophosphate biosynthesis, as demonstrated here), or the biosynthesis of co-factors (e.g. ATP or NADPH) or other global regulators (e.g. ppGpp). Its scalability offers new possibilities for high-throughput flux profiling of a broad range of metabolic (sub)systems, at minimal cost and effort. ScalaFlux can easily be adapted to measure fluxes through other biological processes, such as protein turnover.

Overall, in addition to broadening the range of metabolic systems that can be investigated, ScalaFlux enhances the following aspects of $^{13}$C-MFA: minimal data and analytical requirements (fluxes can be estimated robustly from just a few measurements from the metabolic subsystem of interest, which can typically be collected using a single platform since closely related metabolites often have similar physico-chemical properties); independence from physiology (no need to measure nutrient uptake fluxes, growth rates, or biomass compositions); computational efficiency and stability (smaller equation systems with fewer free parameters); short labeling times (no tracer incorporation required to reach steady state), which allows dynamically changing fluxes to be probed; applicability to rich media (where measuring the many extracellular fluxes and labeling patterns of all the nutrients is difficult and may create computational bottlenecks); and better flux identifiability (because of its intrinsic scalability and robustness to missing measurements and network gaps).

ScalaFlux can be applied alone or in combination with other methods to address a broad range of biological questions. Combined with untargeted MS(/MS) approaches [34–36], ScalaFlux paves the way to $^{13}$C-flux studies at the cellular level. The network coverage of untargeted

MS(/MS) approaches is in general low and sparse, which results in poor flux identifiability when the complete dataset is integrated into metabolic reconstructions. In ScalaFlux, incomplete datasets can still be exploited to estimate fluxes through subsystems, and these flux measurements can be used to constrain genome scale metabolic models. Our approach should also be helpful to study poorly characterized organisms, for which simulations from carbon entry up to the pathway of interest may not be possible.

From a computational point of view, the proposed approach shares many elements with traditional approaches, and is compatible with all current simulation frameworks–elementary metabolite units (EMUs), cumomers, fluxomers, etc—[1,14,25]. The approach introduced here can be implemented in existing $^{13}$C-flux calculation software [10,26,28,37] with minimal effort. Indeed, the key features required to implement ScalaFlux in existing software focus on the definition of label input(s). These features, which are necessary and sufficient to allow the conceptual change brought by ScalaFlux, consist in fitting analytical functions to experimental labeling dynamics of the (local) label inputs, and in using time-dependent functions as label input to model isotope propagation. As proof of concept, we have implemented ScalaFlux in IsoSim, a versatile modeling software designed to integrate proteomics, metabolomics and isotopic data with stoichiometric, kinetic, regulatory and thermodynamic constraints to enhance functional analyses of metabolic systems. We present ScalaFlux to measure fluxes in metabolic (pseudo) steady-state condition. Future development will be geared towards coupling Scala-Flux with kinetic modeling, and thereby offer the possibility of analyzing dynamic fluxome rearrangements.

## Methods

### Implementation of the ScalaFlux workflow

We implemented the ScalaFlux workflow (Fig 1C) in a major update of IsoSim, an R software previously developed to couple kinetic and isotopic models of metabolism [21]. The source code of IsoSim v2 is freely distributed under open-source license at https://github.com/MetaSys-LISBP/IsoSim/. Briefly, IsoSim includes functions to i) construct flux models, ii) design isotope labeling experiments, iii) simulate label propagation, and iv) fit experimental data in order to estimate fluxes. To implement the ScalaFlux approach in IsoSim, we developed novel functions to i) fit the experimental labeling dynamics of the (local) label inputs with analytical functions and ii) use time-dependent functions as label input to model isotope propagation. Each of these steps is explained in detail in the following sections.

All the scripts we used to construct the models, to perform the simulations and to generate the figures are provided at https://github.com/MetaSys-LISBP/IsoSim/ to ensure reproducibility and reusability.

### Construction of flux models

IsoSim requires the following information to construct a flux model: i) the set of reactions of interest, ii) the tracer atom transitions of each reaction, and iii) the accessible isotopic data. IsoSim then automatically constructs the minimal system of ordinary differential equations (ODEs) required to simulate the accessible isotopic measurements. The detailed procedures and algorithms we used to construct the models can be found in the initial article on IsoSim [21], which has been enhanced with the EMU framework [27] to reduce the size of the equation system.

Note that each flux can be defined either as constant or calculated using a kinetic equation which may depend on metabolite concentrations. IsoSim can thereby perform both stoichiometric and kinetic modelling.

## Design of isotope labeling experiments

The present framework provides i) direct identification of the minimal set of label input(s) that need to be measured for a given flux model, and ii) simulations for different configurations (e.g. different pools, flux distributions or local label input dynamics). These two features are crucial to support experimental design and ensure flux identifiability before performing the experiments [22].

## Fitting local label input(s)

The labeling dynamics of all the EMUs identified as local label input(s) must be measured or estimated. IsoSim implements methods to convert these discrete measurements into continuous analytical functions. It is important to note that neither the analytical function nor the estimated parameters have any biological meaning. The aim of this step is just to define a sufficiently accurate representation of the isotopic profiles of the local label input(s).

Experimental $^{13}$C-enrichment dynamics of local label input(s) can be fitted by a logistic function (Eq 1):

$$y(p, t) = \frac{p_1}{1 + e^{-p_2 \cdot (t - p_3)}} \tag{1}$$

where $p$ is the vector of parameters to estimate (here $p_1$, $p_2$ and $p_3$), and $t$ is time. We also implemented a double-logistic function (Eq 2) to fit more complex labeling dynamics, as proposed by Elmore et al. [38]:

$$y(p, t) = p_4 + (p_5 - p_6 \cdot t) \cdot \left( \frac{1}{1 + e^{(p_7 - t)/p_8}} - \frac{1}{1 + e^{(p_9 - t)/p_{10}}} \right) \tag{2}$$

Parameter estimation is formulated as a constrained non-linear optimization problem (Eq 3):

$$minimize\ f(p)$$

$$subject\ to\ g(p) > c \tag{3}$$

where $p$ is the parameter vector, $f$ is the objective function that evaluates the deviation between the simulated and measured data, $g(p)$ is the constraint function, and $c$ is the constraint vector. The objective function $f$ (Eq 4) is defined as the sum of squared weighted errors:

$$f(p) = \sum_i \left( \frac{x_i - y_i(p, t_i)}{\sigma_i} \right)^2 \tag{4}$$

where $x_i$ is the experimental value of data point $i$ collected at time $t_i$, with an experimental standard deviation $\sigma_i$, and $y_i(p, t_i)$ is the corresponding simulated value. Constraints are defined for all parameters to be estimated ($0 < p_1 < 1$, $-100 < p_2 < 100$, $-1000 < p_3 < 1000$ for the logistic function and $-1 < p_4 < 1$, $-1 < p_5 < 1$, $-10 < p_6 < 10$, $-1000 < p_7 < 1000$, $-100 < p_8 < 100$, $-1000 < p_9 < 1000$, $-100 < p_{10} < 100$ for the double logistic function) to improve convergence by reducing the solution space. The optimization problem is first solved using particle swarm optimization (R 3.2.4, *pso* package v1.0.3), followed by an L-BFGS-B [39] search (with an upper limit of 1000 iterations) to improve convergence. A plot of measured versus fitted data is generated to allow visual inspection of the quality of fit, and the analytical functions describing local label inputs are provided as output.

## Simulation of label propagation

IsoSim solves the ODE system to simulate label propagation through the metabolic subnetwork of interest, using as input i) the constructed model, ii) the analytical functions describing local label input(s), iii) the metabolite concentrations, and iv) the fluxes. The simulation engine is based on the fluxomer framework [25], as detailed in [21], and has been enhanced using the EMU framework [27]. This facilitates the identifiability analysis while significantly reducing the size of the equation system to be solved.

## $^{13}$C-flux calculation and sensitivity analysis

Fluxes are estimated by fitting experimental data (the concentrations and labeling dynamics of metabolic intermediates). The objective function $h$ (Eq 5) is defined as the sum of squared weighted errors [40]:

$$h(v, m) = \sum_i \left( \frac{x_i - y_i(v, m)}{\sigma_i} \right)^2 + \sum_j \left( \frac{n_j - m_j}{\sigma_j} \right)^2 \qquad (5)$$

where $v$ is the vector of fluxes, $m$ is the vector of metabolite concentrations $m_j$, $x_i$ is the experimental value of the labeling at data point $i$, with experimental standard deviation $\sigma_i$, $y_i(v,m)$ is the corresponding simulated value, $n_j$ is the experimental concentration of metabolite $m_j$ with standard deviation $\sigma_j$. Equality and inequality constraints can be defined for the fluxes (default constraints: $-10^3 < v < 10^3$) and metabolite concentrations (default constraints: $10^{-6} < m < 10^3$). The objective function $h$ is minimized using the *nlsic* optimization algorithm [10] (with 50 iterations). The goodness-of-fit is evaluated using a chi-square test. Finally, we applied a Monte-Carlo sensitivity analysis [29] to estimate the flux precision. This non-linear method of sensitivity analysis consists in i) generating noisy datasets (according to the experimental standard deviations of the measured concentrations and labeling of metabolites), ii) calculating fluxes for each of these synthetic datasets, and iii) quantifying the flux uncertainty (i.e. mean, median, standard deviation and 95% confidence intervals) from the spread of the estimated fluxes.

## Supporting information

**S1 Table. Initial values of fluxes and metabolite concentrations.** Values of fluxes and metabolite concentrations used to simulate label propagation through the example network shown in Fig 1A.
(XLSX)

**S1 Fig. Simulation results.** Simulated labeling dynamics of all metabolites of the example network (Fig 1A) in response to a switch from unlabeled $X_{out}$ to fully labeled $X_{out}$, for fluxes and metabolite concentrations given in S1 Table.
(PDF)

**S2 Fig. Fit of local label inputs.** The labeling dynamics of the local label inputs of all the subsystems shown in Fig 3 were fitted with analytical functions (as detailed in the Methods section), based on the simulation results given in S1 Fig. The dots represent the fitted data and the lines represent the best fits.
(PDF)

**S3 Fig. Flux calculation results.** For all minimal subsystems of the example network (Fig 3), fluxes were estimated by fitting the labeling dynamics of the metabolic intermediate(s), using

as (local) label input(s) the analytical functions obtained from the fits given in S2 Fig. For each subsystem, the dots represent the fitted data and the lines represent the best fits. The flux values and confidence intervals estimated from these fits are shown in Fig 3.
(PDF)

**S4 Fig. Impact of the quality of the fit of label inputs on the estimated fluxes.** For two minimal subsystems ($S_N$ and $S_F$ in panels A and B, respectively), we degraded artificially the quality of the fit of label inputs by varying parameters of the analytical functions (100 sets of parameters were randomly sampled within ± 5% of their optimal values), and we calculated how the fluxes estimated from the degraded analytical functions deviate from the true values. Plots show the error on the estimated fluxes (and on the relative contribution of the two converging reactions for $S_F$) as function of the error on the representation of label inputs (sum of squared residuals for the degraded analytical functions of label inputs). The red dots represent results for the best fits (i.e. with parameters of the analytical functions set to their optimal values).
(PDF)

## Acknowledgments

We thank MetaboHUB-MetaToul (Metabolomics & Fluxomics Facilities, Toulouse, France, www.metatoul.fr), which is part of the MetaboHUB-ANR-11-INBS-0010 national infrastructure (www.metabohub.fr), for providing free access to its computational resources. We thank Matthieu Guionnet for technical assistance and Baudoin Delépine for insightful comments on the manuscript.

## Author Contributions

**Conceptualization:** Pierre Millard, Uwe Schmitt, Patrick Kiefer, Julia A. Vorholt, Stéphanie Heux, Jean-Charles Portais.

**Funding acquisition:** Julia A. Vorholt, Jean-Charles Portais.

**Investigation:** Pierre Millard, Uwe Schmitt, Patrick Kiefer, Jean-Charles Portais.

**Methodology:** Pierre Millard, Uwe Schmitt, Patrick Kiefer, Jean-Charles Portais.

**Software:** Pierre Millard.

**Supervision:** Jean-Charles Portais.

**Visualization:** Pierre Millard.

**Writing – original draft:** Pierre Millard.

**Writing – review & editing:** Pierre Millard, Uwe Schmitt, Patrick Kiefer, Julia A. Vorholt, Stéphanie Heux, Jean-Charles Portais.

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
