## [Editor Report · Decision Letter 0]

18 Sep 2019

Dear Dr Portais,

Thank you very much for submitting your manuscript 'ScalaFlux: a scalable approach to quantify fluxes in metabolic subnetworks' for review by PLOS Computational Biology. We note that you submitted this as a Research Article under the category of 'Systems Biology', but we feel that this is much better suited for our Methods section, and would like to handle it as such. Therefore, if you are happy to reusbmit your paper as Methods Article, with the correct formatting and requirements, we encourage you to do so. More information regarding the Methods section can be read here: https://journals.plos.org/ploscompbiol/s/submission-guidelines#loc-methods-submissions

Please note, the criteria of substance of methodical advance, thorough comparative validation and accessibility must be met.

Please do the let the Journal Office know if you have any questions about this.

Sincerely,

Thomas Lengauer

Methods Editor

PLOS Computational Biology

---

## [Decision Letter · Decision Letter 1]

20 Jan 2020

Dear Dr. Portais,

Thank you very much for submitting your manuscript "ScalaFlux: a scalable approach to quantify fluxes in metabolic subnetworks" for consideration at PLOS Computational Biology.

As with all papers reviewed by the journal, your manuscript was reviewed by members of the editorial board and by several independent reviewers. In light of the reviews (below this email), we would like to invite the resubmission of a significantly-revised version that takes into account the reviewers' comments.

We cannot make any decision about publication until we have seen the revised manuscript and your response to the reviewers' comments. Your revised manuscript is also likely to be sent to reviewers for further evaluation.

Sincerely,

Kiran Raosaheb Patil, Ph.D.

Associate Editor

PLOS Computational Biology

Thomas Lengauer

Methods Editor

PLOS Computational Biology

Reviewer's Responses to Questions

**Comments to the Authors:**

Reviewer #1: The authors present a novel method for 13C-flux analysis with clear advantages over current approaches. By defining suitable subnetworks, flux estimation is possible with ScalaFlux for pathways not accessible with current methods aiming for holistic flux fitting. While I find the advantages very important and interesting and the method generally well presented, I think that some issues require clarification. Please, find my detailed concerns below.

1. The authors state that ScalaFlux has been implemented on top of previously developed methods in software IsoSim. It remains somewhat unclear what are the novel parts that are proposed here for publication. Please, clarify the novelty compared to previous methods in IsoSim.

2. It remains also not clearly stated, for what kind of growth conditions is the ScalaFlux method suitable to. It should be clearly stated that metabolic (pseudo) steady state is required.

3. It does not also become clear how the limitations of mean molecular enrichment data are reflected in the flux identifiability or unidentifiability in specific kinds of subnetworks involving cleavage

4. In addition, it remains unclear how does the shape of the function fitted to convert the discrete data into continuous affect the flux estimation.

5. In Introduction on line 70 it is stated that the intracellular concentrations of metabolites would need to be measured for flux identifiability, but this of course does not hold for many steady state 13C-flux analysis methods as the above in the Introduction mentioned metabolic flux ratio analysis.

6. Flux units are missing from page 12 and from figures 3 and 4.

Reviewer #2: Major

The approach was mostly demonstrated over an example network. The real system used is rather a simple linear-like pathway. I believe the approach should be applied on one other real system with a larger pathway to better demonstrate its applicability.

Minor

1) The abbreviation EMU is not defined in the manuscript. Please shortly define it.

2) The pipeline in Figure 1 includes sensitivity (Monte Carlo analysis). However this step is never defined in the manuscript. To ensure thoroughness of the paper it should shortly be defined/explained in the manuscript.

3) In Figure 1, D is formed from Xin, but in Fig.2 and Fig.3 It is typed as Sin, not Xin. Please correct. Similarly, in Supplementary Table, I see Sin and Sout, not defined in Figure 1. Probably they should be Xin and Xout. Also, the flux v6xch in the supplementary table is not defined in Figure 1.

4) Supplementary figures are a bit confusing .Please give more information in the legends of supplementary figures. It also looks supplementary figure labels are mixed. In supplementary figure S1, what do the figure titles (S1, S4_9 etc) correspond to? And why do we see a red and black curve in S12 in this figure?

5) Regarding Figure 3A and Supplementary Table 1, Is there any specific reason behind selecting and specifying those five fluxes out of all fluxes?

6) The following explanation in line225-226 is not clear: “assuming a typical precision 0.02 for 13C-enrichments and of 10 % for concentrations”. What is precision? Why is one defined as absolute value and the other one as a percentage?

**Have all data underlying the figures and results presented in the manuscript been provided?**

Reviewer #1: Yes

Reviewer #2: Yes

PLOS authors have the option to publish the peer review history of their article (what does this mean?). If published, this will include your full peer review and any attached files.

Reviewer #1: No

Reviewer #2: No
---

## [Decision Letter · Decision Letter 2]

19 Mar 2020

Dear Dr. Portais,

We are pleased to inform you that your manuscript 'ScalaFlux: a scalable approach to quantify fluxes in metabolic subnetworks' has been provisionally accepted for publication in PLOS Computational Biology.

Best regards,

Kiran Raosaheb Patil, Ph.D.

Associate Editor

PLOS Computational Biology

Thomas Lengauer

Methods Editor

PLOS Computational Biology

Reviewer's Responses to Questions

**Comments to the Authors:**

Reviewer #1: All my concerns were well addressed by the Authors' revision.

Reviewer #2: The authors adressed my comments properly.

**Have all data underlying the figures and results presented in the manuscript been provided?**

Reviewer #1: Yes

Reviewer #2: Yes

PLOS authors have the option to publish the peer review history of their article (what does this mean?). If published, this will include your full peer review and any attached files.

Reviewer #1: No

Reviewer #2: No

---

## [Editor Report · Acceptance letter]

7 Apr 2020

PCOMPBIOL-D-19-01486R2 

ScalaFlux: A scalable approach to quantify fluxes in metabolic subnetworks

Dear Dr Portais,

I am pleased to inform you that your manuscript has been formally accepted for publication in PLOS Computational Biology. Your manuscript is now with our production department and you will be notified of the publication date in due course.

With kind regards,

Sarah Hammond
